# Artificial Intelligence-Based Opportunities in Liver Pathology—A Systematic Review

**DOI:** 10.3390/diagnostics13101799

**Published:** 2023-05-19

**Authors:** Pierre Allaume, Noémie Rabilloud, Bruno Turlin, Edouard Bardou-Jacquet, Olivier Loréal, Julien Calderaro, Zine-Eddine Khene, Oscar Acosta, Renaud De Crevoisier, Nathalie Rioux-Leclercq, Thierry Pecot, Solène-Florence Kammerer-Jacquet

**Affiliations:** 1Department of Pathology CHU de Rennes, Rennes 1 University, Pontchaillou Hospital, 2 rue Henri Le Guilloux, CEDEX 09, 35033 Rennes, France; 2Impact TEAM, Laboratoire Traitement du Signal et de l’Image (LTSI) INSERM, Rennes 1 University, Pontchaillou Hospital, 35033 Rennes, France; 3Research Unit n°UMR1341 NuMeCan–Nutrition, Métabolismes et Cancer, Rennes 1 University, Pontchaillou Hospital, 2 rue Henri Le Guilloux, CEDEX 09, 35033 Rennes, France; 4Department of Liver Diseases CHU de Rennes, Rennes 1 University, Pontchaillou Hospital, 35033 Rennes, France; 5Assistance Publique-Hôpitaux de Paris, Department of Pathology Henri Mondor, 94000 Créteil, France; 6INSERM U955, Team Pathophysiology and Therapy of Chronic Viral Hepatitis and Related Cancers, 94000 Créteil, France; 7Department of Urology, CHU de Rennes, Rennes 1 University, Pontchaillou Hospital, 2 rue Henri Le Guilloux, CEDEX 09, 35033 Rennes, France; 8Department of Radiotherapy, Centre Eugène Marquis, 35033 Rennes, France; 9Biosit Platform UAR 3480 CNRS US18 INSERM U955, Rennes 1 University, Pontchaillou Hospital, 35033 Rennes, France

**Keywords:** digital pathology, liver, hepatology, deep learning, artificial intelligence, performance metrics

## Abstract

Background: Artificial Intelligence (AI)-based Deep Neural Networks (DNNs) can handle a wide range of applications in image analysis, ranging from automated segmentation to diagnostic and prediction. As such, they have revolutionized healthcare, including in the liver pathology field. Objective: The present study aims to provide a systematic review of applications and performances provided by DNN algorithms in liver pathology throughout the Pubmed and Embase databases up to December 2022, for tumoral, metabolic and inflammatory fields. Results: 42 articles were selected and fully reviewed. Each article was evaluated through the Quality Assessment of Diagnostic Accuracy Studies (QUADAS-2) tool, highlighting their risks of bias. Conclusions: DNN-based models are well represented in the field of liver pathology, and their applications are diverse. Most studies, however, presented at least one domain with a high risk of bias according to the QUADAS-2 tool. Hence, DNN models in liver pathology present future opportunities and persistent limitations. To our knowledge, this review is the first one solely focused on DNN-based applications in liver pathology, and to evaluate their bias through the lens of the QUADAS2 tool.

## 1. Introduction

### 1.1. Introduction to AI

Artificial Intelligence (AI) is probably the next revolution in pathology. “Artificial Intelligence” is a broad term encompassing many different approaches for problem-solving. Among them, Machine Learning (ML), and Deep Neural Networks (DNN) in particular, are capable of handling huge amounts of data by increasingly complex mathematical systems that have proven time and time again to at least perform similarly to pathologists for the segmentation of clinically relevant regions and classification of tumors. Moreover, they might contribute to standardizing care by reducing inter-observer lack of reproducibility. Inspired by neuroanatomy, algorithms known as Deep Learning Algorithms (DLA) rely on architectures known as Deep Neural Networks (DNN) and are able to produce astounding results in imagery recognition and classification. As a mostly image-based discipline, pathology is particularly suitable for the application of such AIs. Therefore, there is a growing field of interest, both academic and industrial, for the development of efficient, fast, robust and precise AI-based solutions for pathologists, among them liver pathology [1].

### 1.2. Principles of Deep Neural Networks Algorithms

In pathology, most DNN-based tasks can be separated into three categories: segmentation, classification and prediction [2]. Briefly, segmentation algorithms aim to identify anatomical structures on a virtual slide. For example, they may be useful for detecting cell boundaries, tumoral areas or structures such as portal tracts or fibrotic septa [3,4]. Classification algorithms aim to discriminate between, in general, binary modalities, “tumoral vs. non tumoral” being the most common example. Non-binary modalities can also be investigated through the lens of artificial intelligence, some examples in liver pathology being fibrosis grading and hepatitis severity assessment. Prediction algorithms function similarly to classification algorithms, taking input data and proposing an output answer. The difference between classification and prediction algorithms lies in how the “ground truth” is established; in a classification model, “ground truth” can be established directly from the input data (in pathology: a Whole Slide Image), whereas in a prediction model other techniques are to be performed (such as molecular testing). Moreover, “prediction” models include a temporal aspect, because their predicted outcome will not be measurable immediately (either because additional analysis is required, or because clinical follow-up is needed).

Notwithstanding their assigned goal, all DNN-based algorithms share some similarities [5]. DNN systems can be either supervised (Supervised Learning SL) or unsupervised (UL). SL requires the pre-definition of a “ground truth” through the annotations of the developmental dataset. In UL, the model is intended to find a pattern by itself through the numerous unannotated examples it is given, so that it is generalizable to the data ensemble. Development of a DNN-based algorithm is divided into training, testing and validation phases. On the training dataset, the algorithm is given inputs (annotated example of interest if the model is supervised) so as to determine the optimal combinations of features giving the most correct outputs. The validation step aims to fine-tune the hyper-parameters of the algorithm (for example, the number of artificial neuron layers) and to prevent overfitting on the training dataset. Finally, the testing dataset evaluates the model’s performance metrics. In order to obtain high precision, a huge amount of data is often required. Moreover, in supervised learning, the dataset must be labeled to establish a “ground truth” for training the model. The SL can be either “supervised”, with many refined annotations made by an expert (for example, manual definition of precise Region Of Interest for a segmentation model) or “semi-supervised” (for example, a “tumor” or “non-tumor” label for a classification model). Hence, establishing a sufficiently large and labeled dataset is time-consuming and may introduce bias, such as lack of reproducibility by experts [6]. Some data augmentation principles (mathematical transformations of images such as artificial blurring, rotations and inversions, changes of color...) have been proposed and are commonly used to try and mitigate such issues [5].

### 1.3. Evaluation of Algorithms by Performance Metrics

Throughout the literature, many performance metrics have been used to evaluate DNN algorithms. What they refer to and how they compare to one another is sometimes challenging. The most commonly used performance metrics and their major characteristics are summarized in Table 1. A 2 × 2 contingency table may be used to either present or explicit the metrics; such an example is given in Figure 1. It is to be noted that no performance metric is widely recognized as superior to the others, nor are they comparable to one another, and as such they are often used in conjunction. In ideal conditions, performance assessment is to be performed on an external validation set, so as to generalize the results and not risk a bias of overfitting the algorithm to the developmental dataset and causing an inability to generalize performance with unseen data (staining differences, artefacts, etc.) [7].

### 1.4. Aim of the Present Review

As Artificial Intelligence in healthcare is an ever-growing subject, many publications have aimed to review its impact in numerous medical fields, be it radiology, pathology or clinical specialties [8]. The present study aims to provide a synthetic and comprehensive review of applications provided by DNN algorithms in liver pathology throughout the literature, in the tumoral as well as metabolic and inflammatory fields, and to discuss the potential bias and opportunities opened by Artificial Intelligence for liver pathologists.

## 2. Materials and Methods

This systematic review follows the Preferred Reporting Items for Systematic Review and Meta-Analyses (PRISMA) statement (Appendix A) [9].

### 2.1. Inclusion and Exclusion Criteria

All studies published until 12 January 2022 using deep learning in hepatology on histopathologic slides are included. Exclusion criteria were as follows: (1) not using Whole Slide Images (WSI) of human tissue slides; (2) not published in English; (3) review articles, editorials or other unrelated topics.

### 2.2. Online Registration Information

The search protocol was not registered online prior to data extraction.

### 2.3. Data Sources and Literature Search Strategy

The first author (A.P.) searched the Pubmed and Embase databases to identify studies up to 01 December 2022. The search terms used were as follows: (Liver OR Hepatic) AND (artificial intelligence OR deep learning OR machine learning) AND (Whole slide image OR digital pathology OR pathomics OR pathomic) AND (Human). Manual selection of relevant articles through crosschecking references was additionally performed.

### 2.4. Studies Selection and Data Extraction

The first author screened the articles according to inclusion/exclusion criteria, removed duplicates and added relevant articles through crosschecking references. For each article, information regarding authors, year of publication, country, algorithm type (segmentation vs. classification vs. prediction), number of WSI used for development and validation, and the most pertinent performance outcomes were extracted. One author (A.P.) extracted data from each study and a second independent author (K-J.S-F.) validated the extracted data. All studies were separated and summarized in two tables: “tumoral” for all studies related to cancer segmentation, classification or prediction (Table 2), and “non-tumoral” for all other liver pathologies (Table 3). The features extracted included author and year of publication, first author’s country, category of algorithm, size of developmental and external validation sets, and main performance metrics as appropriate.

### 2.5. Assessment of the Risk of Bias and Applicability

Each article was evaluated keeping the Quality Assessment of Diagnostic Accuracy Studies (QUADAS-2) guidelines in mind, so that any potential bias and limitations could be discussed [10]. Notably, the following criteria were taken into account to stratify as “high risk”: patient selection: developmental set <60 WSI, unicentric study; index test: no external validation set, no use of image augmentation principles; reference standard: definition of “ground truth” solely based on one expert, no use of standardized scoring system when appropriate, lightly-supervised or unsupervised model; flow and timing: for prospective studies only, patient exclusions from final analysis, multiple reference standard provided.

## 3. Results

### 3.1. Search Results

The result of the search yielded 167 articles (127 non-duplicated). After excluding articles based on title and abstract screening, 35 articles remained and were reviewed by full text screening. Seven articles were added manually by manual reference checking, amounting to 42 selected articles for systematic review. Twenty-two articles were labeled “tumoral” and 20 articles were labeled “non-tumoral” based on full text review (Figure 2).

### 3.2. Tumoral

Twenty-two articles were reviewed through full text screening (Table 2) [11,12,13,14,15,16,17,18,19,20,21,22,23,24,25,26,27,28,29,30,31,32]. No studies used prospectively collected data. The algorithms primary goal was either segmentation (6/22) [11,12,13,14,15,16], classification (6/22) [17,18,19,20,21,22] or prediction (10/22) [23,24,25,26,27,28,29,30,31,32]. The majority of studies (18/22) were focused on hepatocellular carcinoma. The others were respectively reporting on liver metastasis (3/22) [12,19,28] or cholangiocarcinoma (1/22) [32]. Six studies used The Cancer Genome Atlas (TCGA) database for either their developmental or external validation set (including the one from Liao et al., whose full-text access was unavailable through our institution [25]). More than half of the studies (13/22) tested their algorithms on at least one external validation set. The DNN models used in each study are reported in Appendix A.

Three published segmentation studies used the same developmental and external testing set, namely the Pathology Artificial Intelligence Platform (PAIP) dataset from the PAIP 2019 Challenge [11,13,14]. PAIP challenge’s goal is to evaluate new and existing algorithms for automated liver cancer detection in WSI, either with or without peritumoral/intratumoral stroma-reaction, using a predefined fully annotated dataset from Seoul National University Hospital, South Korea [33]. As of the writing date of the present article, more than 1500 participating teams were numbered, and the PAIP 2019 Challenge is still ongoing. Among the published studies, Roy et al. demonstrated the highest performance, using a classification model to achieve an F1-score of 0.94 [13].

Cheng et al. proposed a DNN-classification model to help discriminate hepatic nodular lesions (HNL), comprising hepatocellular adenoma, hepatocellular carcinoma, dysplatic nodule and focal nodular hyperplasia [22]. Their model was trained and validated on HE biopsy specimens and showed a good discrimination power between (pre)neoplastic lesions and benign hepatocellular proliferation or background with an AUC of 0.94.

DNN-prediction models were shown to be able to predict genetic somatic mutations or signatures directly from HE WSI. Chen et al. demonstrated an algorithm able to predict CTNNB1, FMN2, TP53 and ZFX4 somatic status in HCC slides ailing from TCGA, with AUCs ranging from 0.71 to 0.89 [23]. Zeng et al. developed a DNN-model predicting activation of 6 immune gene signatures in advanced HCC directly from histology [30]. Those immune gene signatures were previously shown to be associated with better response/survival rates after nivolumab therapy (an anti PD-1 antibody).

Prediction of post-surgical recurrence-free interval or survival in HCC was also explored through DNN-based systems. Saillard et al. proposed a DNN-based system computing a risk score of independent prognostic value after hepatocellular surgery based on the resected tumor’s WSI (c-index 0.70) [25]. That score was shown to be significant on an external validation dataset, even after stratification of patients on the disease’s stage, vascular invasion and presence of satellite nodules, and to outperform a composite score based on clinical, biological and pathological features. In the same vein, Yamashita et al. developed a DNN approach of post-surgical recurrence risk stratification for HCC with WSI from the TCGA and validated their model on the Stanford Department of Pathology slide archive, achieving an AUC of 0.96 and a c-index of 0.67 on the external validation set with a 10-year follow-up.

Among the individual studies, the highest amount of WSI used for the developmental dataset was 2917 WSI in a predictive study from Chen et al. [29]. This study aimed to predict the existence of microvascular invasion in patients with HCC by a DNN analysis of the WSI on the primary tumor. Their model was deemed able to successfully and independently predict microvascular invasion in the external validation set with an AUC of 0.87. Furthermore, Chen et al. strived to implement their model into a clinical setting and showed that the presence of microvascular invasion as predicted by their DNN model is correlated with a poorer outcome.

**Table 2 diagnostics-13-01799-t002:** Key features of reported studies regarding the use of Deep Neural Networks (DNN) on Whole Image histology Slides (WSI) in the “tumoral” field.

Authors (Year) [Reference]	Country	Algorithm Goal	Development Dataset	External Validation Dataset	Performance Metrics
Feng Y. et al. (2021) [11]	France	segmentation (HCC)	60 WSI	40 WSI	Jaccard score 0.90 F1 score 0.47
Cancian P. et al. (2021) [12]	Italy	segmentation (LM)	303 WSI	no	Jaccard score 0.89
Roy M. et al. (2021) [13]	USA	segmentation (HCC)	60 WSI	40 WSI	F1 score 0.94
Wang X. et al. (2021) [14]	China	segmentation (HCC)	60 WSI	40 WSI	Jaccard score 0.797
Feng S. et al. (2021) [15]	China	segmentation (HCC)	592 WSI	157 WSI (TCGA)	Accuracy 0.88
Yang TL. et al. (2022) [16]	Taiwan	segmentation (HCC)	46 WSI	no	Jaccard score 0.89
Li S. et al. (2017) [17]	China	segmentation (HCC)	127 WSI	not provided	Accuracy 0.97 F1 score 0.94
Kiani A. et al. (2020) [18]	USA	classification (HCC and CCK)	70 WSI	80 WSI	Accuracy 0.84
Schau GF. et al. (2020) [19]	USA	classification (LM)	257 WSI	no	F1 score 0.77
Ercan C. et al. (2022) [20]	Switzerland	classification (HCC)	98 WSI	no	Accuracy 0.84 F1 score 0.91
Diao S. et al. (2022) [21]	China	classification (HCC)	100 WSI (TCGA)	no	AUC 0.92
Cheng N. et al. (2022) [22]	China	classification (HCC)	649 WSI	234 WSI	AUC 0.94
Chen M. et al. (2020) [23]	China	prediction (HCC)	387 WSI	101 WSI	AUC 0.71
Liao H. et al. (2020) [24]	China	prediction (HCC)	not provided	not provided	AUC 0.89
Saillard C. et al. (2020) [25]	France	prediction (HCC)	390 WSI	342 WSI (TCGA)	c-index 0.70
Yamashita et al. (2021) [26]	USA	prediction (HCC)	299 WSI (TCGA)	198 WSI	c-index 0.67
Saito A. et al. (2021) [27]	Japan	prediction (HCC)	158 WSI	no	Accuracy 0.90
Xiao C. et al. (2022) [28]	China	prediction (LM)	611 WSI	no	AUC 0.85
Chen Q. et al. (2022) [29]	China	prediction (HCC)	2917 WSI	504 WSI	AUC 0.87
Zeng Q. et al. (2022) [30]	France	prediction (HCC)	349 WSI (TCGA)	139 WSI	AUC 0.92
Qu WF. et al. (2022) [31]	China	prediction (HCC)	576 WSI	147 WSI (TCGA)	c-index 0.71
Xie J. et al. (2022) [32]	China	prediction (CCK)	766 WSI	no	AUC 0.68

HCC: Hepatocellular Carcinoma; CCK: Cholangiocarcinoma; LM: Liver Metastasis.

### 3.3. Non-Tumoral

Twenty articles were reviewed through full text screening (Table 3) [34,35,36,37,38,39,40,41,42,43,44,45,46,47,48,49,50,51,52,53]. As for tumoral studies, none used prospectively collected data. The algorithms primary goal was either segmentation (4/20) [34,35,36,37], classification (15/20) [38,39,40,41,42,43,44,45,46,47,48,49,50,51,52] or prediction (1/20) [53]. The most prevalent thematic included inflammation detection and quantification, especially in Non-Alcoholic SteatoHepatitis (NASH) (6/20) and fibrosis detection and grading (4/20). Two studies reported on the evaluation of steatosis to evaluate the quality of the graft in liver transplantation [36,48]. All studies relied on HE staining, with or without additional immunohistochemical stains, except for one, which relied only on CK7 immunochemistry [50]. No study validated their algorithm on an external validation dataset, be it designed for either segmentation, classification or prediction. The DNN models used in each study are reported in Appendix A.

The number of WSI incorporated into the developmental set was disparate among studies. In particular, Puri developed a machine learning model on 1277 digital slides to detect drug-induced liver injuries on the cellular level, based on both rat and human hepatocytes, achieving 0.99 Accuracy [43]. On the other end of the spectrum, studies such as Gawrieh et al. and Perez Sans et al. showed the feasibility of developing a steatosis classifier on a training dataset as low as 18 and 20 WSI, respectively [47,48].

NASH scoring is a combination of multiple cardinal features, namely steatosis, lobular inflammation, hepatocyte ballooning and fibrosis. Attempts to develop automated DNN-based classification algorithms for NASH scoring occurred as early as 2015 by Vanderbeck et al., whose model achieved an AUC of 0.95 for lobular inflammation and 0.98 for hepatocyte ballooning detection [39]. Other teams used machine learning programs to help discriminate between NAFLD and NASH [45,46], chronic hepatitis and NASH [49] or alcoholic steatohepatitis and NASH [51]. Heinemann et al. proposed an algorithm not only capable of identifying the cardinal features of NASH but also able to score NASH as a discrete pathologist-like score (as per the Kleiner score of NASH activity score, NAS) [52]. Inter-observator agreement between the DNN algorithm and pathologists was best for steatosis and fibrosis class assessment (kappa 0.92 and 0.81, respectively), but dropped for ballooning and inflammation class assessment (kappa 0.42 and 0.40, respectively).

Constantinescu et al. were the sole proposer of a predictive deep-learning based model to predict early (within 30 days) post-surgical complications after liver surgery [53]. To that end, their model assessed neoangiogenesis and inflammation through both HE and immunohistochemical stains and was able to predict the arising of complications with an accuracy of 0.97 and an AUC of 0.97 on the internal testing dataset.

**Table 3 diagnostics-13-01799-t003:** Key features of reported studies regarding the use of Deep Neural Networks (DNN) on Whole Image histology Slides (WSI) in the “non tumoral” field.

Authors (Year) [Reference]	Country	Algorithm Goal	Development Dataset	External Validation Dataset	Performance Metrics
Guo X. et al. (2018) [34]	USA	segmentation (steatosis)	451 WSI	no	Jaccard score 0.77 F1 score 0.66
Jirik M. et al. (2020) [35]	Czech Republic	segmentation (intra vs. extralobular)	33 WSI	no	Accuracy 0.91
Roy M. et al. (2020) [36]	USA	segmentation (steatosis)	36 WSI	no	F1 score 0.94
Yu H. et al. (2022) [37]	USA	segmentation (portal tracts)	53 WSI	no	Jaccard score 0.80 F1 score 0.89
Vanderbeck S. et al. (2014) [38]	USA	classification (steatosis, bile ducts, vascular structures)	47 WSI	no	AUC 0.83
Vanderbeck S. et al. (2015) [39]	USA	classification (NASH)	59 WSI	no	AUC 0.98
Wang TH et al. (2015) [40]	Taiwan	classification (fibrosis)	175 WSI	no	AUC 0.82
Munsterman I. et al. (2019) [41]	Netherlands	classification (NASH)	79 WSI	no	AUC 0.97
Klimov S. et al. (2019) [42]	USA	classification (fibrosis)	115 WSI	no	AUC 0.70
Puri M. (2020) [43]	USA	classification (DILI)	1277 WSI	no	Accuracy 0.99
Forlano et al. (2020) [44]	UK	classification (NASH)	246 WSI	no	AUC 0.80
Teramoto T. et al. (2020) [45]	Japan	classification (NASH)	79 WSI	no	AUC 0.85
Salvi M. et al. (2020) [46]	Italy	classification (steatosis)	385 WSI	no	Accuracy 0.97
Gawrieh S. et al. (2020) [47]	USA	classification (NASH)	18 WSI	no	AUC 0.79
Perez-Sans F. et al. (2021) [48]	Spain	classification (steatosis)	20 WSI	no	AUC 0.98
Marti-Aguado D. et al. (2021) [49]	Spain	Classification (chronic hepatitis)	156 WSI	no	AUC 0.75 (NASH) AUC 0.99 (Chronic Hepatitis model)
Sjöblom N. et al. (2021) [50]	Finland	classification (chronic cholestatis)	210 WSI	no	Accuracy 0.93
Ramkissoon R. et al. (2022) [51]	USA	classification (NASH)	97 WSI	no	AUC 0.96
Heinemann F. et al. (2022) [52]	USA	classification (NASH)	467 WSI	no	F1 score 0.37 to 0.85
Constantinescu C. et al. (2022) [53]	Romania	prediction (liver surgery complications)	500 WSI	no	AUC 0.97

NASH: Non-Alcoholic Steatohepatitis; NAFLD: Non-Alcoholic Fatty Liver Disease; DILI: Drug Induced Liver Injuries.

### 3.4. Assessment of the Risk of Bias and Applicability through the QUADAS-2 Tool

The overall assessment of the reviewed studies is presented in Figure 3. When the overall risk of bias was measured with the QUADAS-2 tool, a majority of studies (30/42) presented at least one high-risk factor. Algorithms developed on unicentric datasets inferior to 60 WSI were deemed at high risk for the patient selection domain. All studies without an external validation set were judged at high risk for the “index test domain”. For the “reference standard” domain, high risk was applied if “ground truth” was determined by only one pathologist and on criteria deemed either too subjective or that were notoriously low-reproducible. Finally, as no study fitted the criteria for evaluation by the “flow and timing” domain, it was categorized as “not applicable”. Per study evaluation with the QUADAS-2 tool is provided in Appendix A.

## 4. Discussion

As shown, the applications and opportunities offered by AI are manifold, even for a field as specialized as liver pathology, in terms of both the tumoral (hepatocellular carcinoma, cholangiocarcinoma and secondary tumors in liver) and non-tumoral (NAFLD and NASH, chronic hepatitis, fibrosis staging, iatrogenic-induced lesions) aspects. DNN-based models are such promising tools that they have led to numerous publications throughout the last decade, with an increasing rate over the past few years. We have strived to provide a comprehensive, systematic and updated review of AI in liver pathology. Pathologists may sometimes feel estranged regarding this topic, because AI is foremost a mathematical and computational discipline, with a wide array of performance metrics, architectures and other specificities. Hence, navigating the literature might sometimes prove challenging. Previous reviews have discussed the opportunities of AI in liver diseases as a whole, or as part of clinical, radiological and sometimes pathological applications in gastroenterology and hepatology [2,3,4,5,6,7,8]. To our knowledge, this review is the first to solely focus on DNN-based applications in liver pathology, and to evaluate their bias through the lens of the QUADAS2 tool.

### 4.1. AI in Tumoral Liver Pathology: What to Remember

The practical value of segmentation algorithms in the tumoral liver pathology field may seem poor, because most of the time pathologists have no trouble identifying HCC areas in the liver. In a similar fashion, classification algorithms such as Kiani et al.’s [18], which aimed to distinguish HCC and CCK, may seem of limited value, because there are already robust immunohistochemical tools available. However, the major contribution of such algorithms might be a gaining of time in everyday practice, and one might argue that this alone is pertinent enough in the context of a shortage in liver pathologists, as has been raised by several authors in other domains of pathology [54,55,56,57]. The development of classification algorithms targeting lesions of difficult diagnosis such as benign hepatocellular proliferations (hepatocellular adenoma and focal nodular hyperplasia) might prove the most beneficial, especially on biopsy specimens, such as in Chens et al.’s study [22].

The development of DNN-based algorithms tackling the tumoral field in liver pathology seem to be more advanced and robust throughout the literature than those in the non-tumoral field. Indeed, our global evaluation through the QUADAS-2 tool showed that studies in the tumoral field were less prone to risk of bias in all domains. The reasons for this include the availability of public online databases such as The Cancer Genome Atlas (TCGA), which compiles molecular data and whole slide images of numerous cancers, which are an invaluable resource for the development and external validation of many DNN-based algorithms for pathology, including hepatocellular carcinoma [15,26,32]. Likewise, multicentric studies with a development cohort in one hospital and an external testing dataset in another may be easier to design for DNN-based studies in the tumoral rather than the metabolic field [22,24,30].

The most promising uses for DNN-based algorithms in tumoral hepatology seem to lie in prediction algorithms. Predicting recurrence [23] or survival [26] from histological WSI is especially appealing, because its clinical applications in patient’s care are obvious and, as of today, out of reach. The prediction of gene signature without resorting to molecular testing [31] could also prove a gain of both time and money. As of yet, there is no targeted therapy in HCC. The prediction of genomic alterations in CCK might, on the other hand, yield therapeutical value [58,59]. One must keep in mind that the predictive information of survival or recurrence is complex and most certainly multimodal. As such, DNN-based algorithms could provide yet another tool for patient management, in association with existing pathological (TNM classification), clinical (Performance Status) or radiology scoring systems [60,61]. Hence, multimodal approaches such as DNN-based systems compiling radiological and pathological imaging data (reflected through the concepts of “radiomics” and “pathomics”, respectively) have been proposed for non-small lung cancer [62], cervical cancer [63], breast cancer [64] and glioblastoma [65] but have yet to be developed in liver pathology. Radiological imaging data are particularly suited to the development and use of deep learning artificial intelligence and share many of its concepts and limitations with the field of pathology [66].

### 4.2. AI in Non-Tumoral Liver Pathology: What to Remember

On the other hand, classification DNN-based systems are of paramount interest in the metabolic liver pathology field. For example, steatohepatitis diagnosis and grading has been shown to present poor inter-observer reproducibility [67]. In particular, hepatocyte ballooning recognition seems to be poorly reproducible, even between expert pathologists, hindering DNN development on the subject, as acknowledged by authors such as Brunt et al. [68]. DNN-based models may prove useful in the future to standardize such difficult diagnosis, yet a “reference standard bias” may represent a pitfall in their development if overlooked. To avoid this, “ground truth” (i.e., the annotations) should be defined by standard scoring systems such as the Non-Alcholic Fatty Liver Disease Activity Score for NAFLD and by a consensus of more than one pathologist, and this could be built once again on a multimodal level including clinical, biological and radiological data.

While external validation is feasible enough for studies on tumoral material (13/22 in this review), it seems an acute limitation for those on inflammatory or metabolic diseases. In fact, among the papers we reviewed, none (0/20) of them tested their model on an external validation set. There are in fact, at the time of this article’s writing, no online database similar to TCGA for metabolic or inflammatory liver diseases. Setting up such a public annotated database might prove beneficial to put DNN-based algorithms to trial.

### 4.3. Persistent Limitations to the Implementation of AI in Daily Practice

AI integration into healthcare increases at an ever-growing speed, and DNN-based models seem to be promising tools in the image analysis field as a whole, in radiology and, since the arising of whole slide imaging, in pathology. At present, image-analysis DNN-tools do not outmatch trained physicians, but they might prove useful for alleviating complex and time-consuming tasks [69]. As great as their promises may be, DNN systems are hampered by some limitations. Most of the studies reported in this review were stratified as having at least one domain of high risk of bias according to the QUADAS-2 tool. Some steps should be taken when designing a DNN-based study, among them (but not limited to) building a multicentric, sufficiently large development database with a “ground truth” defined by consensual, if possible multimodal, criteria and systematically validated on an external dataset. However, economic and practical constraints should be taken into consideration, for most DNN models require a curated and sufficiently large developmental dataset, which can prove time-consuming and costly [70]. The explainability of predictive algorithms is also a common shortcoming that some authors have begun to tackle by looking back at the most predictive histological tiles of their models [25,29,30]. To further that end, the role of pathologists will be of paramount importance. Ethical questions such as minorities and low-income country integration should also always be kept in mind [71], because AI’s usefulness relies on its success in a clinical setting, and its scope is always increasing [72,73]. As such, predictive risk-stratification models in particular may be of the utmost interest.

### 4.4. Present Review’s Limitations

We focused on two databases, namely Pubmed and Embase, and may have missed papers published in specialized, non-healthcare-related journals. It is also to be noted that the QUADAS-2 tool, though encouraged by the PRISMA guidelines, is not perfectly suited for the qualitative assessment of AI-related studies. Indeed, the QUADAS-2 guidelines do not provide precise criteria to assess the robustness of AI-related studies. Our criteria stratifying “low risk of bias” and “high risk of bias” are therefore subjective. QUADAS-IA is an extension of QUADAS-2 and is specifically designed to tackle this issue, and to date is still to be published [74]. When available, it should prove an even more precise tool to evaluate diagnostic research in the AI field. Cabitza and Campagner also proposed a checklist of 30 items for the qualitative assessment of medical AI studies [75]. While not specifically designed for pathology, it may be useful when designing an AI-based study in healthcare; it is available online and was additionally performed in the present review for each study (Appendix A) [76].

## 5. Conclusions

For the moment, contrary to other fields of pathology, there are no commercially available AI-based solutions in liver pathology. Studies are still impaired by a lack of external validation cohorts. Therefore, multicentric studies and expert collaborations are needed to further develop DNN-based algorithms.

## Figures and Tables

**Figure 1 diagnostics-13-01799-f001:**
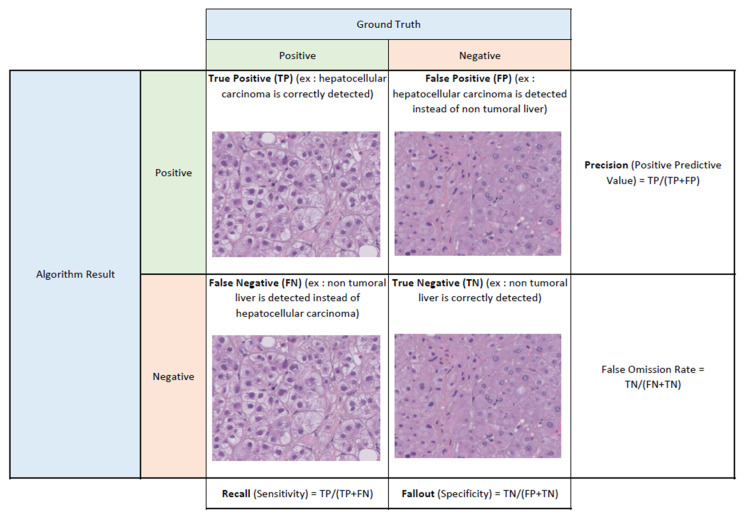
Example of a 2 × 2 contingency table used for calculating performance metrics such as precision, recall and fallout. Histological images are courtesy of the CHU Pontchaillou, Rennes, France.

**Figure 2 diagnostics-13-01799-f002:**
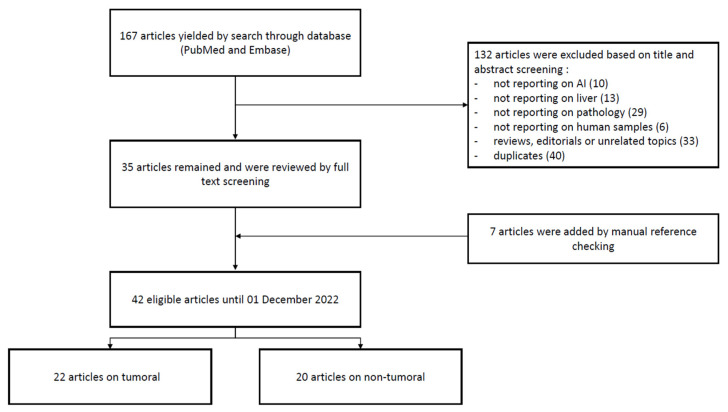
Flowchart of literature search.

**Figure 3 diagnostics-13-01799-f003:**
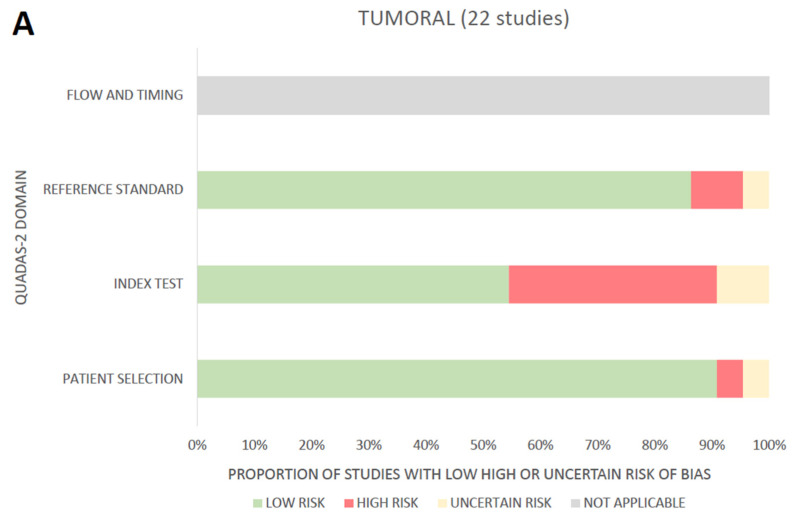
Global overview of reviewed studies, including evaluation of risk of bias through the QUADAS-2 tool for the “tumoral” (**A**) and “non tumoral” (**B**) fields.

**Table 1 diagnostics-13-01799-t001:** Most commonly used performance metrics used in the literature, with key features summarized.

Performance Metric	Rationnal	Interpretation	Key Concepts	Main Applications
Jaccard Index	Also known as the ratio of Intersection over Union (IoU)	0 ≤ x ≤ 1 the closer to 1 the best	Very simple to use, the Jaccard index is a way of conceptualizing accuracy for object detection. It quantifies the similarity of the algorithm vision with those of the annotated ground truth	object/area segmentation (may also be employed for binary classification)
Accuracy	Number of correct predictions (true positives and true negatives) divided by the total number of predictions	0 ≤ x ≤ 1 the closer to 1 the best	Very simple to use, accuracy quantifies the percentage of correct predictions by the algorithm. However, it is not adapted to imbalanced problems (where positive and negative proportions are greatly different) nor to problems where the “cost” of false positives/negatives must be taken into account (such as screening situations). Therefore, it may not be suited for evaluating algorithms in certain medical situations	classification/prediction
F1-score	Combines Precision (ratio of true positives/total positives predicted) and Recall (ratio of true positives/total positives in ground truth)	0 ≤ x ≤ 1 the closer to 1 the best enhancing Precision OR Recall leads to a better score	Useful for evaluating performance in situations where Accuracy would be misleading (see above). It is adapted to unbalanced problems. It is, however, difficult to interpret when low: is it because of low Precision (too much false positives) or low Recall (not enough true positives)?	classification/prediction
AUROC (or AUC)	Combines Recall (ratio of true positives/total positives in ground truth) and Fallout (ratio of false positives/total negatives in ground truth)	0 ≤ x≤1 the closer to 1 the best enhancing Precision OR diminishing Fallout leads to a better score	Useful for evaluating the diagnostic ability of a (binary) classifier, because it takes both true positives and true negatives into account. Therefore (and contrary to F1-score), diminishing the number of false negatives is taken into account (which is of importance in screening situations)	classification/prediction
c-index (concordance index)	Generalization of AUROC for assessing the correct ranking of events	0 ≤ x ≤ 1 the closer to 1 the best	Adapted to datasets with censored data (survival studies, prediction of adverse events, etc.)	

## Data Availability

The data presented in this study are available in the Appendix A.

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
