# Peer review of "Artificial Intelligence-Based Opportunities in Liver Pathology—A Systematic Review"

_diagnostics, 2023, doi:10.3390/diagnostics13101799_

Round 1

Reviewer 1 Report

This is a well prepared and fairly comprehensive review of liver pathology and AI. It made an interesting read, with a nice brief introduction to AI methods and performance metrics. The figures were well made and the tables were easy to understand. I did notice some publications on liver and AI were not included in this review, which was recognized by the authors in the limitations section of the manuscript (some articles not accessible through Embase and Pubmed were missed). 

Reviewer 2 Report

This paper summarizes the research results of artificial intelligence (AI) deep neural networks (DNNs) in the field of liver pathology, and the result indicates that DNN-based models are well-represented in the field of liver pathology. However, the manuscript still contains many points and needs to be revised. 

  1. In the abstract, the GAP needs to be proposed. Is it because the predecessors did not summarize this aspect? Or is the summary of the predecessors not systematic enough? Or a different perspective? 
  2. Several pieces of literature should be added in the revised form to summarize the use of AI in medicine, such as: 

https://doi.org/10.1016/j.bspc.2023.104641; 

10.2352/J.ImagingSci.Technol.2023.67.3.030402; 

https://doi.org/10.3390/app13063489; 

https://doi.org/10.1148/radiol.2020192224;

10.1109/CBMS49503.2020.00111;

https://doi.org/10.1016/j.ejmp.2021.02.006.

  1. Using only the PubMed and Embase databases and lots of related literature will be missed. Why not use the Web of Science? 
  2. It is recommended to include each specific model name in the table and mark the data that achieves the best results.
  3. It is suggested to provide a flowchart of the model that provides the best results, if possible.
  4. Assessment of the Risk of Bias and Applicability through the QUADAS-2 tool, please provide some specific data and describe the results. 

Minor editing of the English language is required, such as DNN-algorithms in line 118.

Reviewer 3 Report

In this article, authors present a review on publications where Deep Neural Networks handle
several applications in image analysis. Authors present in a detailed way relevant aspects and limitations of the reviewed publications, such as if external validation is present.

Some IA/ML aspects are described in this article in a unclear or confusing way. These must be  corrected or completed.  

Line 50 - “...such as Artificial Neural Networks (ANN) or Deep Neural Networks
(DNN) and are…”  DNN is a type do ANN, so use just one description, or rephrase. Otherwise it seems those ae two disjoint concepts.  So chose either ANN or DNN.

Line 60 – The methods/approaches to segmentation are inumerous. Thus, the reference to “region-based segmentation, edge-based segmentation or mixed methods” it is too narrow, and clearly useless. So either remove it or else it should replaced by a more complete description of segmentation methods.  

Line 64 – Are there examples of severity level classification in liver pathology ? Along with binary classification ?

Line 66 -  In the very large and always growing AI communities (with different technical backgrounds) there are in fact several definitions that are not coherent.   Concerning “classification” and “prediction” the authors refer to the “semantic” difference between both. Actually, in many scientific  areas “prediction” should include the time aspect. So it would be a classification refering to some time in the future relatively to the available data.  This aspect should be clarified in the article.

Line – 80 – instead of “attune” it is preferably “tune”   or “fine-tune” and “parameters should be replaced by “hyper-parameters”  (since training it also tune “parameters”)

line 87 – do not confuse a more complete annotation versus a simpler one, with “weakly supervised” which amounts to a semi-supervised approach that can be compared with a “supervised” one.

Table1 F1-score weakeness –  Although this may be true in some cases, in general there is no need to define a threshold.
Although table 1 is a summary view, once, there are several performance metrics in the reviewed works, and the numbers in different metrics are not comparable, there should be a more complete explanation about the definition, and the intuitive meaning of those metrics. These is important specially if the audience is not from the AI/ML domain.     

Finally, QUADAS-IA is an extension of QUADAS-2 and would be the right tool to analyze the reviewed works.  While QUADAS-IA is not published,  there are several “checklists” for the application of IA in the medical area e.g. TRUE (is it “True? Reproducible? Is It Useful? Is It Explainable?”), but several other exist, one such list is cited in page 14. These tools put the focus on specific aspects of IA and its application would have been of great value.  I recommend the one such tool should be applied.

Round 2

Reviewer 2 Report

The manuscript has been improved a lot according to the reviewers' comments. The authors carefully checked the whole manuscript and addressed all the comments seriously. It is suggested that the manuscript can be accepted in the present form.